



# Temperature dependent sensitivity of iodide chemical ionization mass spectrometers

Michael A. Robinson[1,2,3], J. Andrew Neuman[1,2], L. Gregory Huey[4], James M. Roberts[1], Steven S. Brown[1,3], and Patrick R. Veres[1]

[1]NOAA Chemical Sciences Laboratory, Boulder, Colorado, USA
[2]Cooperative Institute for Research in Environmental Sciences (CIRES), University of Colorado Boulder, Boulder, Colorado, USA
[3]Department of Chemistry, University of Colorado Boulder, Boulder, Colorado, USA
[4]School of Earth and Atmospheric Science, Georgia Institute of Technology, Atlanta, GA, USA

*Correspondence to*: Michael A. Robinson (michael.a.robinson@noaa.gov) and Patrick R. Veres (patrick.veres@noaa.gov)

**Abstract**

Iodide chemical ionization mass spectrometry (CIMS) is a common analytical tool used in both laboratory and field experiments to measure a large suite of atmospherically relevant compounds. Here, we describe a systematic ion molecule reactor (IMR) temperature dependence of iodide CIMS analyte sensitivity for a wide range of analytes in laboratory

experiments. Weakly bound iodide clusters such as HCl, HONO, HCOOH, HCN, phenol, 2-nitrophenol and acyl peroxy nitrate (PAN) detected via the peroxy radical cluster, all exhibit strong IMR temperature dependence of sensitivity ranging from -3.4 to 5.9 % °C$^{-1}$ (from 37 to 47 °C). Strongly bound iodide clusters such as $Br_2$, $N_2O_5$, $ClNO_2$, and PAN detected via the carboxylate anion all exhibit little to no IMR temperature dependence ranging from 0.2 to -0.9 % °C$^{-1}$ (from 37 to 47 °C). The IMR temperature relationships of weakly bound clusters provide an estimate of net reaction enthalpy and comparison with

database values, indicating these clusters are in thermal equilibrium. Ground site HCOOH data collected in the summer of 2021 in Pasadena, CA are corrected showing a reversal in the diel cycle, emphasizing the importance of this correction (35 ± 6 % during the day, -26 ± 2 % at night). Finally, we recommend two approaches to minimizing this effect in the field, namely heating or cooling the IMR; the latter has the added benefit of improving absolute sensitivity and reducing drift in harsh field environments.

**1 Introduction**

Atmospheric traces gases participate in chemical reactions that impact air quality and climate. Quantification of these trace gases, often at parts per trillion levels, demands highly sensitive and selective measurements. For decades, chemical ionization mass spectrometers (CIMS) have been used to provide rapid and sensitive in situ measurements of a wide variety of trace gases and aerosol composition. More recently, time of flight mass spectrometers (ToF-MS) have been developed to measure tens or

even hundreds of compounds simultaneously, making their field deployment increasingly common in the last decade (Bertram et al., 2011; Lee et al., 2014; Lopez-Hilfiker et al., 2014; Krechmer et al., 2018; Koss et al., 2018; Lopez-Hilfiker et al., 2019;





Veres et al., 2021). CIMS instruments employ reagent ions to react selectively with analytes of interest, and a number of negative ($I^-$, $Br^-$, $CF_3O^-$, $SF_6^-$, $CH_3C(O)O^-$) and positive ($H_3O^+$, $NO^+$, $NH_4^+$) reagent ions have been utilized in field and lab experiments to quantify a wide range of atmospheric trace gases (Slusher et al., 2004; Crounse et al., 2006; Huey et al., 1995;
Veres et al., 2008; de Gouw and Warneke, 2007; Koss et al., 2016; Zaytsev et al., 2019). Negative ion-adduct chemistry has been used with quadrupole mass spectrometers that sample the atmosphere since 1975 (Dougherty et al., 1975). The use of iodide-adduct ion chemistry has become commonplace in atmospheric chemistry field campaigns as this chemistry has provided sensitive measurements of molecular halogens, halogen oxides, carboxylic acids, acyl peroxynitrates (PANs), oxygenated organics and nitrogen oxides (Slusher et al., 2004; Osthoff et al., 2008; Kercher et al., 2009; Thornton et al., 2010;
Mielke et al., 2011; Riedel et al., 2012; Neuman et al., 2016).

Often iodide ions (and/or its water clusters) react with a neutral analyte molecule to form an adduct that can be identified and detected in a ToF mass spectrometer. Iodide's large mass defect, low likelihood of electron transfer or proton abstraction often allow selective ionization chemistry that can achieve low detection limits and avoid interferences. Additionally, iodide's single natural isotope greatly simplifies complex mass spectral interpretation. Iodide adduct formation is typically only slightly
exergonic (i.e. $\Delta G < 0$), meaning adduct formation is rapid (i.e. every collision forms a cluster ion), and relatively fragment free (i.e. soft ionization) (de Hoffmann and Stroobant, 2007). A complication of iodide CIMS, in contrast to proton transfer reactions, is that sensitivities to different analytes range several orders of magnitude (Iyer et al., 2016) which implies that product ion abundance is not solely determined by forward reaction rates. It has been proposed that this range of sensitivities is due to the variation in binding enthalpy between the iodide anion and neutrals (Iyer et al., 2016).
The iodide anion–molecule chemistry in the presence of water is a system of reactions:

$$I^- + A \rightleftharpoons [I \cdot A]^- \qquad \text{(R1)}$$

$$[I \cdot H_2O]^- + A \rightleftharpoons [I \cdot A]^- + H_2O \qquad \text{(R2)}$$

$$I^- + RC(O)O_2 \rightarrow RC(O)O^- + IO \qquad \text{(R3)}$$

Here, $I^-$ and $I^- \cdot H_2O$ are reagent ions, A is the analyte of interest, and $I \cdot A^-$ is a product ion. Reaction R3 pertains to the special
analyte case of peroxy acetyl radicals, which differs from most analyte cluster reaction pathways (i.e. R1 or R2). The portion of each reaction pathway depends on the analyte and reagent ($[I \cdot H_2O^-]$ or $[I^-]$). Ambient mixing ratios of a given analyte are obtained using Eq. 1:

$$[A] = \frac{[I^-]}{[I \cdot A^-]kt} \qquad \text{(1)}$$

Where $k$ is the effective rate constant of the reaction and $t$ is the reaction time, and this equation also holds true for other
reagent ions (i.e. $I^- \cdot H_2O$). Typically $k$ and $t$ are not determined separately, but the product of the two parameters, or the effective sensitivity, is determined for a given analyte (Huey, 2007).

Much has been done to characterize this ion chemistry for application to atmospheric chemistry field and laboratory studies (Huey et al., 1995; Slusher et al., 2004; Kercher et al., 2009; Lee et al., 2014; Lopez-Hilfiker et al., 2016). Analyte ionization occurs subsequent to mixing with reagent ions in a volume known as the ion molecule reactor (IMR), see Figure 1. Instrument



analyte detection sensitivity is impacted by the effective reaction rate constants and pathways of the analyte and reagent ion(s) in the IMR (R1-R3). Water vapor plays an important role in the iodide-adduct ionization efficiency, which in turn impacts instrument sensitivity (Lee et al., 2014; Slusher et al., 2004; Huey et al., 1995; Kercher et al., 2009). Instrument sensitivity for a given analyte molecule can exhibit a strong to no relationship with water vapor, and this water dependence requires careful calibration of these instruments as a function of IMR water vapor pressure (Lee et al., 2014). This instrument sensitivity

dependence on water has motivated efforts to actively control water or carefully calibrate as a function of water in order to stabilize or determine sensitivity during field deployment (Lee et al., 2014; Slusher et al., 2004; Veres et al., 2021). IMR temperature determines the importance of kinetics versus thermodynamics for determining the outcome of ion molecule reactions, yet there is limited discussion of the importance of IMR temperature in CIMS systems.

    Reactions in the IMR can be in a kinetic or thermodynamic control regime, where the kinetic regime is described as a

sufficiently exergonic reaction (i.e. $\Delta G << 0$) with a slow reverse reaction rate, such that equilibrium is not established in the IMR. The thermodynamic control regime is described as moderately exergonic reaction (i.e. $\Delta G < 0$) with reverse reaction rates ~5 times smaller than the residence time in the IMR allowing the ion-analyte system to reach equilibrium on this timescale. The $SiF_5^-(HNO_3)$ system achieves equilibrium on timescales equivalent to the residence time in most IMRs used in field deployable CIMS (~100 ms) (Huey, 2007). A CIMS using $Br^-$ reagent ions to measure the $Br^-(HO_2)$ cluster exhibited

thermal equilibrium behavior, with a decrease in instrument sensitivity of 20% with an increase of IMR temperature from 20 to 40 °C (Sanchez et al., 2016). Similar thermochemistry systems (i.e. similar $\Delta G$) are likely to equilibrate on similar timescales.

    Limited kinetic studies have been conducted for iodide anion – molecule reactions. Examples include the $N_2O_5$ clustering reaction (R1) ($k = 1.3 \times 10^{-9}\ cm^3\ s^{-1}$ at 293 K) and the carboxylate anion reaction (R3) ($k = 4 \times 10^{-10}$ to $16 \times 10^{-10}\ cm^3\ s^{-1}$ at 298

K) (Huey et al., 1995, p.6; Villalta and Howard, 1996). However, quantum chemical calculations predict collision rates of neutral molecules with $I^-$ for a variety of atmospherically relevant molecules to range from $2 - 4 \times 10^{-9}\ cm^3\ s^{-1}$ (Iyer et al., 2016). The discrepancy between quantum chemical calculations and laboratory studies is likely due to instrument ion transmission. If an analyte is in the thermal controlled regime, the temperature dependent equilibrium constant of this system of reactions will affect the instrument sensitivity. A system in equilibrium is described by the linear form of the Van't Hoff

equation which relates the change in equilibrium constant ($K_{eq}$) of the ion-molecule reaction to the absolute temperature (T):

$$\ln K_{eq} = -\frac{\Delta_r H^0}{RT} + \frac{\Delta_r S^0}{R} \tag{2}$$

where $\Delta_r H^0$ is the standard reaction enthalpy, $\Delta_r S^0$ is the standard reaction entropy, and R is the gas constant. The slope and intercept of the Van't Hoff plot ($\ln K_{eq}$ vs 1/T) gives the enthalpy and entropy change, respectively (Keesee and Castleman, Jr., 1986).

A recent framework for understanding iodide cluster binding energies uses a mass spectrometer ion lens voltage scanning procedure to estimate the relative binding energies of ion adducts (Lopez-Hilfiker et al., 2016). While this approach gives important insight to the stability of ion adducts, it does not directly provide absolute sensitivities. A drawback of the voltage



scanning determination of analyte sensitivity is the impact of voltage scanning on instrument ion transmission. Analyte sensitivity is a product of the net rate of product ion formation and the instrument transmission of those product ions. Additionally, product ion formation dependence on temperature may make these voltage scanning determinations difficult to interpret.

Temperature affects the abundance of product ions, and hence the sensitivity of iodide adduct CIMS, as has been shown for some analytes studied in the laboratory (Villalta and Howard, 1996). We extend these findings to a range of atmospheric trace gases and provide a framework for understanding variations of detection sensitivity with temperature. Here we present a systematic study of IMR temperature dependence of $I^-$ ion molecule reactions for a number of atmospherically relevant analytes. Additionally we estimate thermally controlled reaction pathways in our IMR system by comparing observed reaction enthalpies to literature values.

Previously, several CIMS IMR temperature control methods have been implemented. Documented instrument deployments range from actively heating the IMR region (50 to 60 °C), such as for the Filter Inlet for Gases and Aerosols instrument (Lopez-Hilfiker et al., 2014), to insulation of the IMR region (Lee et al., 2018), to active cooling of the IMR region (15 °C) (Neuman et al., 2002). However, thermal coupling between the IMR and ToF body, which may be difficult to temperature control in field deployments of CIMS instruments (discussed below), may introduce temperature variations in regions where ion molecule reactions occur. Consequently, we compare our experiments to field observations and recommend temperature control strategies to improve and stabilize sensitivity.

## 2 Experimental Approach

The experiments here are conducted on a modified commercially available ToF-MS (https://www.tofwerk.com/products/api-tof/) that has been used extensively for sampling from ground and airborne platforms (Veres et al., 2021). Figure 1 depicts the front end of the instrument, which differs from the commercially available instrument. The NOAA $I^-$ ToF-CIMS (hereafter NOAA CIMS) utilizes a custom pressure controlled inlet that maintains constant instrument pressures independent of ambient pressure. The NOAA CIMS notably operates at lower IMR pressure (40 mbar, rather than 100 mbar used in many) in order to reduce the impact of secondary chemistry in the IMR instruments (Bertram et al., 2011; Lee et al., 2014). It is worth noting that higher pressure IMR systems will likely be more susceptible to thermal effects due to the increased residence time. The remainder of the ToF has not been modified.





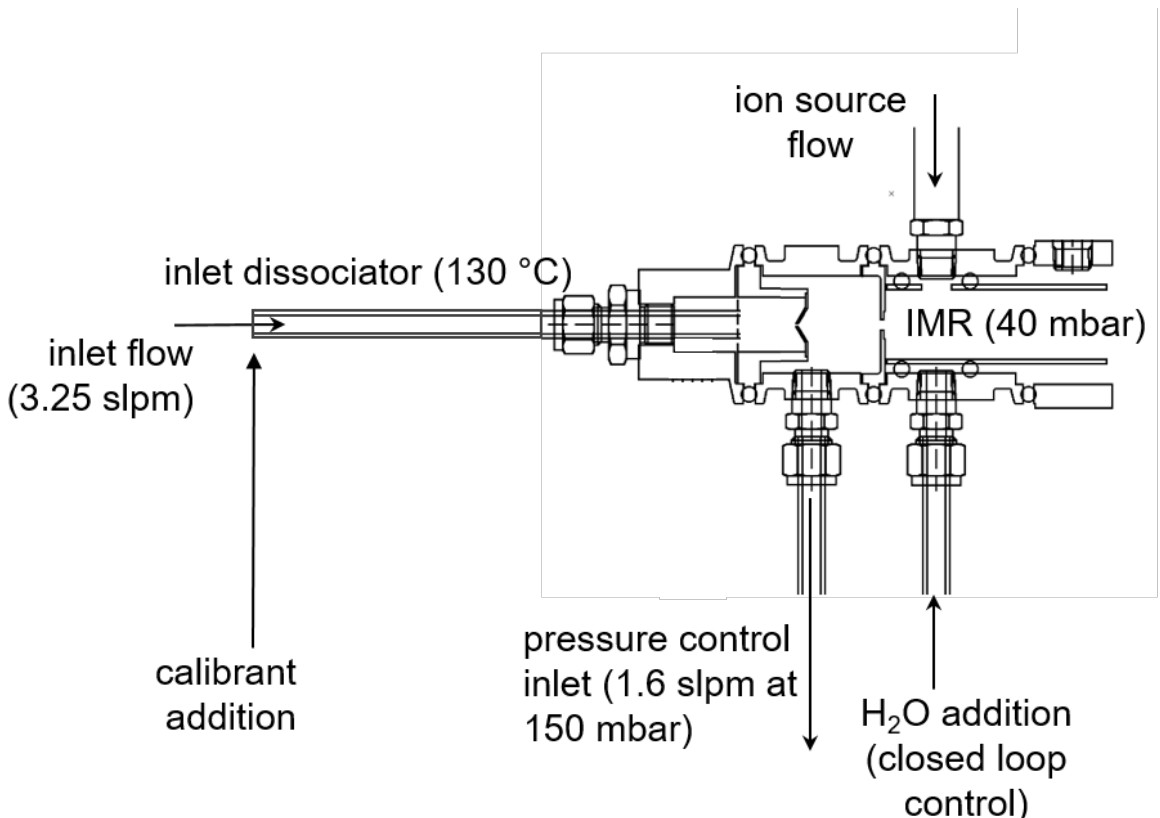

**Figure 1: Instrument inlet schematic.**

The NOAA CIMS detects analyte molecules with $I^-$ and $I^-\bullet H_2O$ ion clusters via ToF-MS with a mass resolving power of approximately 5000 ($\Delta m/m$). Reagent ions are formed by flowing 1 standard liter per minute (slpm) of $N_2$ and 9 standard cubic centimeter per minute (sccm) of 1025 ppm methyl iodide ($CH_3I$ in $N_2$) in front of a vacuum ultraviolet Krypton lamp (Restek photoionization lamp, model 108-BTEX, Restek Corporation, PA, USA). This ionization technique produces an intense and relatively interference-free source of iodide ions (Ji et al., 2020; Breitenlechner et al., 2022). These experiments (unless stated otherwise) were done with water dynamically added to the IMR (via a saturated small $N_2$ flow) to maintain a cluster ratio ($I^-\bullet H_2O:I^-$) (CR) of $0.50 \pm 0.002$, as is typical in field operation of the instrument.

Ambient air was sampled through a mass flow controlled (3.25 slpm) PFA inlet (30 cm length, 0.64 cm ID), which was heated to 130 °C to thermally dissociate PAN for detection via the carboxylate anion (R3) or peroxy radical cluster (Slusher et al., 2004). Ambient air is sampled through a 1000 μm diameter orifice into a pressure controlled region maintained at 150 mbar. This pressure control guarantees that 1.2 slpm enters the IMR, independent of ambient pressure, and mixes with the ~1 slpm flow from the ion source. The reagent ion signals during this work were typically $1\times10^6$ ion counts per second (MHz) for $I^-$ $\bullet H_2O$ and 2 MHz for $I^-$. Product cluster ions are normalized by the $I^-\bullet H_2O$ signal to account for changes in ion source output. Every two minutes, the calibrants are removed and lab air sampled, and the calibration signals are determined as the difference



between lab air alone and lab air with calibrant. Two IMR designs are investigated in this work, the standard Aerodyne Inc, IMR (ARI) and the NOAA IMR, which has been deployed on the instrument for the Atmospheric Tomography – 3 aircraft campaign (AToM – 3, 2017) (https://espo.nasa.gov/atom/archive/browse/atom/id10), AToM – 4 (aircraft campaign, 2018) (https://espo.nasa.gov/atom/archive/browse/atom/id14), Fire Influence on Regional to Global Environments and Air Quality (FIREX-AQ) (aircraft campaign, 2019) (https://www-air.larc.nasa.gov/missions/firex-aq/) and Southwest Urban NOx and VOC experiment (SUNVEx) (ground site, 2021) (https://csl.noaa.gov/projects/sunvex/). Both IMR designs are summarized in Supplemental Table 1.

## 2.2 IMR Temperature experiments

The ion chemistry that forms unique product ions occurs primarily in the IMR. Consequently, the temperature dependence of the sensitivity was predominately examined as a function of IMR temperature. Experiments were conducted using the temperature controllable ARI IMR, which includes a cartridge heater in a stainless steel block mounted to the IMR and controlled via the instrument computer system. All temperatures are obtained from manufacturer supplied thermistors placed on the IMR and ToF. Even without independently heating the IMR, the heat generated by the instrument maintains the IMR temperature at approximately 30 °C. IMR temperature was increased from 30 °C to 45°C over a period of two hours (figure S1). After reaching the maximum IMR temperature, temperature control was turned off to allow the IMR to return to ambient temperature. During the entire experiment, standard additions of analyte molecules occurred every 2 minutes for 45 seconds. Reagent and product ion signals and instrument temperatures during a typical IMR temperature experiment with formic acid (HCOOH) additions is shown in Supplemental Figure 1.

## 2.1 ToF Body temperature experiments

Temperature dependent ion chemistry can occur downstream from the IMR, in which case IMR temperature alone does not determine sensitivity. These regions contain segmented quadrupoles that guide ions, and their temperature is determined largely by the temperature of the entire ToF body, which is closely coupled with ambient temperature. Collisional dissociation occurs during voltage scanning experiments which vary voltages between the first (SSQ) and second (BSQ) quadrupole ion guides, indicating SSQ temperatures may affect sensitivity (Lopez-Hilfiker et al., 2016). For many applications, it may not be feasible to control the temperature of the entire ToF, since it is a very large thermal mass that is often securely mounted when installed on a mobile platform. Consequently, we have investigated the impact of a variable ToF temperature, while controlling IMR temperature, to represent typical field conditions. These additional experiments were conducted using a radiative heater placed next to the ToF body and insulating blanket. Three types of experiments were performed to examine temperature control strategies that could be implemented to stabilize sensitivity if ambient temperatures change substantially. The first is to actively cool the IMR and determine sensitivity while the ambient temperature was varied. This control strategy may afford improved sensitivity for many molecules and reduces sensitivity changes caused by large changes in ambient temperature. The IMR was cooled using a thermoelectric chiller that pumps cold water through copper tubing wrapped around the IMR. This control




strategy achieved an IMR temperature of ~18 °C during these experiments. The second control strategy is to heat the IMR to a temperature higher than the maximum expected ToF body temperature (in this case ~45 °C). This control strategy suffers from a loss in sensitivity for some analytes, but is more easily implemented using the commercially provided hardware compared to cooling the IMR. For both of these active control strategies, the IMR temperature was allowed to stabilize, then the radiative heater was turned on to a set point of ~40 °C, which replicates many harsh field deployment temperatures in aircraft and at ground sites. The third strategy is the absence of active temperature control. This experiment replicates conditions during prior field deployments and reveals the sensitivity changes that can occur in the absence of IMR temperature control. Product ion signals and temperatures during a typical experiment without IMR temperature control is shown in Supplemental Figure 2.

## 2.2 Calibration sources

In order to understand the broader impacts of IMR temperature on detection sensitivity, a wide range of analytes (see Table 1) were added to the instrument while varying the IMR temperature. Iodide ion chemistry detection sensitivity and water dependence varies considerably among analytes (for example Lee et al, 2014, SI). We have tested the IMR temperature dependence for compounds that span a wide range of sensitivities. Table 2 summarizes the calibration method, inlet mixing ratio, reagent-normalized instrument sensitivity (CR = 0.5, IMR temperature = 29 °C) and sensitivity dependence on IMR temperature. Permeation tubes (HCOOH and $Br_2$: Kin-Tek; HCl: VICI Metronics) are held in PFA sleeves and temperature controlled (HCl and $Br_2$ at 30 °C, HCOOH at 40 °C). A continuous 50 sccm flow of $N_2$ is directed around the permeation tubes and through the PFA sleeves. Phenol (99% MilliporeSigma) and 2-Nitrophenol (98% Sigma Aldrich) were added to a glass diffusion cell (Allen Scientific Glass) with a headspace flow of approximately 30 sccm zero air.





| Analyte | Calibration Source | Inlet Mixing ratio | Instrument Sensitivity (Hz pptv$^{-1}$ 1 MHz$^{-1}$ I$^-$•H$_2$O) at CR = 0.50 and IMR temperature = 29 °C | Sensitivity dependence on IMR Temperature (% °C$^{-1}$)[a] |
|---|---|---|---|---|
| HCN | 10.4 ppmv in N$_2$ gas standard | 20 ppbv | 0.05 | -5.7 ± 0.3 |
| HCl | Permeation tube | 27 ppbv | 0.0013 | -5.8 ± 0.4 |
| HCOOH | Permeation tube | 7 ppbv | 5.3 | -5.0 ± 0.2 |
| Br$_2$ | Permeation tube | 1.3 ppbv | 10.8 | -0.79 ± 0.05 |
| ClNO$_2$ | Online synthesis | 30 pptv | 11 | -0.89 ± 0.4 |
| N$_2$O$_5$ | Online synthesis | 3.8 ppbv | 36 | -0.90 ± 0.1 |
| HONO | Online synthesis | 2.5 ppbv | 2.3 | -4.47 ± 0.6 |
| I*PAN | Online synthesis | 800 pptv | 2.4 | -3.41 ± 0.1 |
| PAN Anion | Online synthesis | 800 pptv | 2.1 | 0.2 ± 0.09 |
| 2-Nitrophenol | Diffusion cell | 350 pptv | 0.01 (Palm et al., 2020) | -5.9 ± 0.3 |
| Phenol | Diffusion cell | 1055 pptv | 0.58 (Palm et al., 2020) | -5.4 ± 0.4 |

**Table 1: Calibration sources, instrument sensitivity and IMR temperature dependence of sensitivity for analytes studied.**

**[a]Change in sensitivity from 310 to 320 K**

## 3 Results and Discussion

Most analytes detected in the NOAA CIMS lose sensitivity with increasing temperature, shown in Figure 2 and Table 2. Notably the sensitivity for the PAN anion (R3) does not decrease with temperature and the sensitivity of several halogens

remain nearly constant. Instrument sensitivities are normalized to the sensitivity measured at 310 K, as this allows for a comparison across all field and lab operating conditions examined here. Additionally, normalization permits a comparison of the temperature dependence of analytes with absolute sensitivities (Hz pptv$^{-1}$) that range several orders of magnitude between analytes (Lee et al., 2014; Iyer et al., 2016).



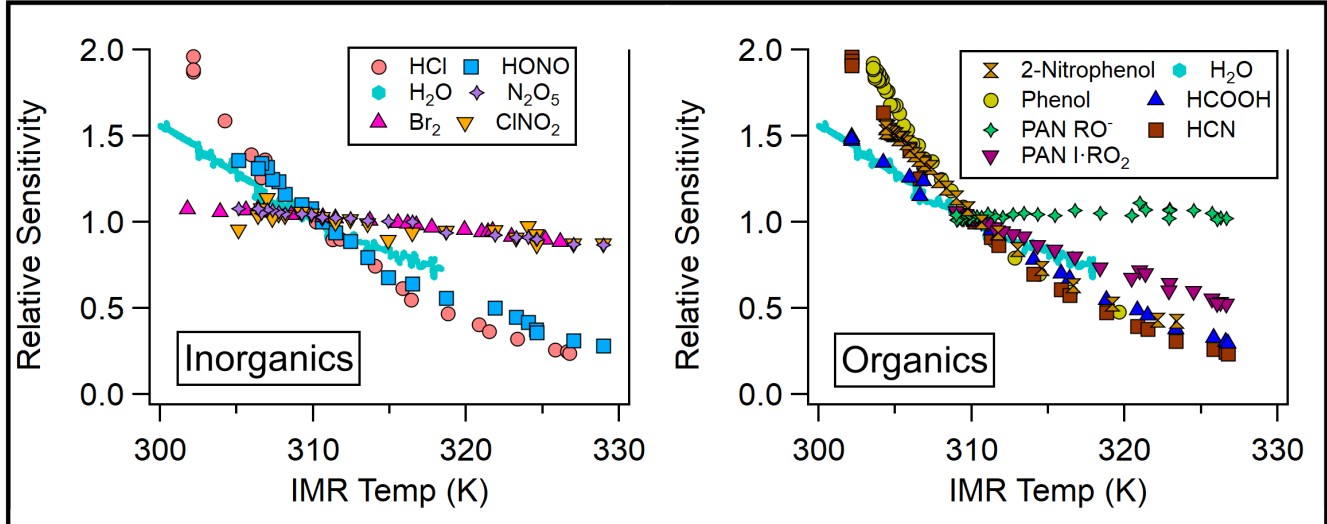

**Figure 2: Relative sensitivity (normalized to sensitivity at 310 K) as a function of IMR temperature for organic and inorganic atmospheric trace gases detected via Iodide adduct chemical ionization.**

The analytes studied here can be grouped into two classes, strongly bound clusters with weak or no IMR temperature dependence (such as $Br_2$, $N_2O_5$, $ClNO_2$, PAN via carboxylate anion detection scheme) and weakly bound clusters with strong IMR temperature dependence (HCl, HONO, HCOOH, HCN and PAN via peroxy radical cluster). The absolute sensitivities of Phenol and 2-Nitrophenol are currently unknown for our instrument, so we have not classified them. However we expect them to have low sensitivity, since these molecules are detected with relatively low sensitivity in other iodide CIMS (0.58 and 0.01 Hz pptv$^{-1}$ respectively) (Palm et al., 2020). Weakly bound inorganic analytes, such as HCl and HONO exhibit reductions of sensitivity of 80% and 67% respectively for an IMR temperature increase from 31 to 50 °C. Weakly bound organic analytes, such as Phenol and HCOOH, exhibit decreases in sensitivity of 75% and 60% respectively when the IMR temperature increases from 30 to 46 °C. Strongly bound inorganic analytes, such as $Br_2$ and $N_2O_5$, exhibit much smaller changes in sensitivity (15%) over these IMR temperature ranges.

**3.1 Impact of IMR heating on water cluster distribution**

IMR heating impacts both product ion formation and the iodide water cluster distribution. We believe that iodide and its first water cluster ($I^-•H_2O$) dominate the reagent ion distribution in our IMR. The detector signal for the most abundant higher order cluster is the second water cluster ($I^-•(H_2O)_2$), which is typically 0.05% of $I^-•H_2O$, and likely reflects the low abundance of higher order clusters in the IMR. IMR temperature sweep experiments were conducted to an IMR temperature of ~55 °C. This is the maximum temperature in which instrument CR could be controlled to 50 % which is the nominal operating CR during field deployment. The reduction in the iodide water cluster product ion as a function of IMR temperature is shown in Figure 2. This experiment was conducted without IMR temperature or water control, in order to understand the influence of IMR





temperature on the $I^-\cdot(H_2O)$ product ion. This thermodynamically controlled behavior of the $I^-\cdot(H_2O)$ product ion emphasizes

the importance of temperature control to maintain a constant reagent ion distribution.

**3.2 IMR temperature dependence gives insight into iodide ion chemistry**

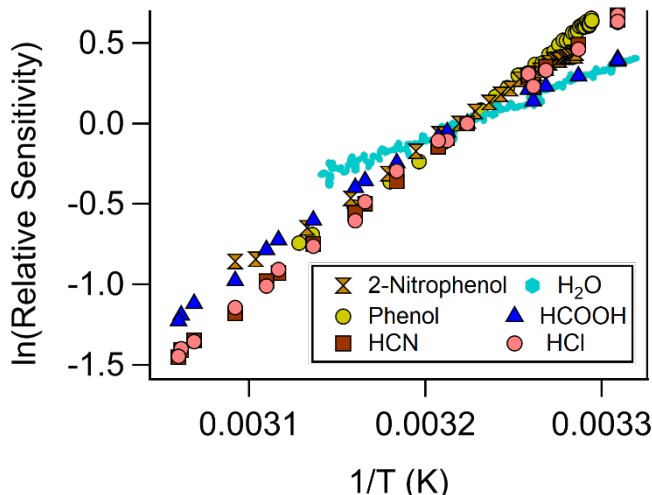

**Figure 3: Van't Hoff relationship for a selection of atmospheric trace gases using Iodide adduct ion chemistry in the NOAA CIMS**
**(normalized at 310 K).**

Analyte systems that equilibrate in the IMR can be investigated further with the Van't Hoff relationship (Keesee and

Castleman, Jr., 1986), shown in Figure 3. The positive slopes indicate that the net reactions of these analytes are exothermic

(i.e. $\Delta_r H < 0$). Analyte systems in equilibrium give a measurement of the net reaction enthalpy in the IMR, according to the

slope of the Van't Hoff relationship in Figure 3. These measured reaction enthalpies can be represented by a combination of

240 the two predominant reaction pathways (R1 & R2) reaction enthalpies. These condition-specific reaction enthalpies are

compared to reported values (https://webbook.nist.gov/cgi/cbook.cgi?ID=C20461545&Units=SI&Mask=40#Ion-Cluster),

shown in Figure 4. In order to compare measured reaction enthalpies to literature values, the ligand switching (R2) reaction

enthalpies must be calculated, by subtracting the single water cluster reaction enthalpy from the analyte direct cluster reaction

enthalpy (blue in Figure 4), an example calculation is shown below for the HCOOH ligand switching reaction:

$$[H_2O \cdot I^-] \rightleftharpoons H_2O + I^- \qquad \Delta_r H^0 = 43 \pm 3 (kJ/mol)$$

$$HCOOH + I^- \rightleftharpoons [HCOOH \cdot I^-] \qquad \Delta_r H^0 = -79.1 \pm 4.2 (kJ/mol)$$

$$HCOOH + [H_2O \cdot I^-] \rightleftharpoons [HCOOH \cdot I^-] + H_2O \qquad \Delta_r H^0 = -79.1 \pm 4.2 + 43 \pm 3 = -36.1 \pm 7.2 \ (kJ/mol)$$





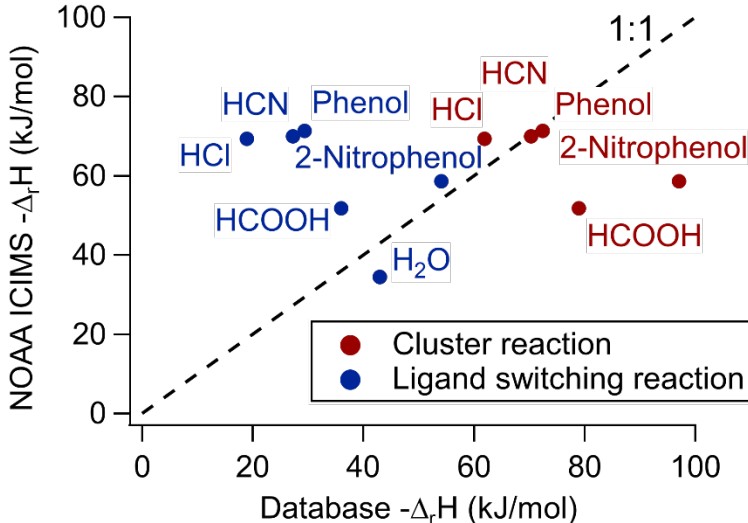

**Figure 4: NOAA CIMS measured reaction enthalpies vs NIST webbook database reaction enthalpies for either the direct clustering pathway or ligand switching pathway. A 1:1 line is included to indicate where the measured reaction enthalpy agrees with the database.**

The direct cluster and ligand switching reaction enthalpies are shown in red and blue, respectively in Figure 4. Where the 1:1 line falls between these values is an indication of the apparent branching ratio between each reaction pathway in the IMR under these conditions. For the case of HCOOH, the net reaction enthalpy is -51.8 ± 1.4 kJ/mol. Consequently, this thermochemical system lies between the ligand switching reaction enthalpy of -36.1 ± 5.1 kJ/mol and the direct cluster reaction enthalpy of -79.1 ± 4.2 kJ/mol. This result, which is only valid for conditions in the IMR during this experiment, implies that HCOOH reacts both through the ligand switching pathway and the direct cluster pathway. This differs from HCN, HCl and phenol, which all fall close to the 1:1 line for the direct clustering pathway (see Figure 4). The involvement of the ligand switching chemistry may explain the relatively high sensitivity for HCOOH as compared to extremely low sensitivity direct clustering analytes, such as HCN, HCl and phenol.

### 3.3 Field deployment and IMR temperature control

The impact on HCOOH sensitivity due to IMR temperature changes during the SUNVEx field campaign are shown in Figure 5. During the day (09:00 to 18:00 Pacific Daylight Time (PDT)), elevated instrument temperatures caused a decrease in HCOOH sensitivity, requiring campaign average correction of 35 ± 6 % (average ± 1 standard deviation). At night (20:00 to 06:00 PDT), instrument temperatures were lower than lab determined sensitivities increasing HCOOH sensitivity, requiring campaign average correction of -26 ± 2 %. The diel dependence of HCOOH is reversed if this temperature correction is not applied making it critical to accurate interpretation of these data.





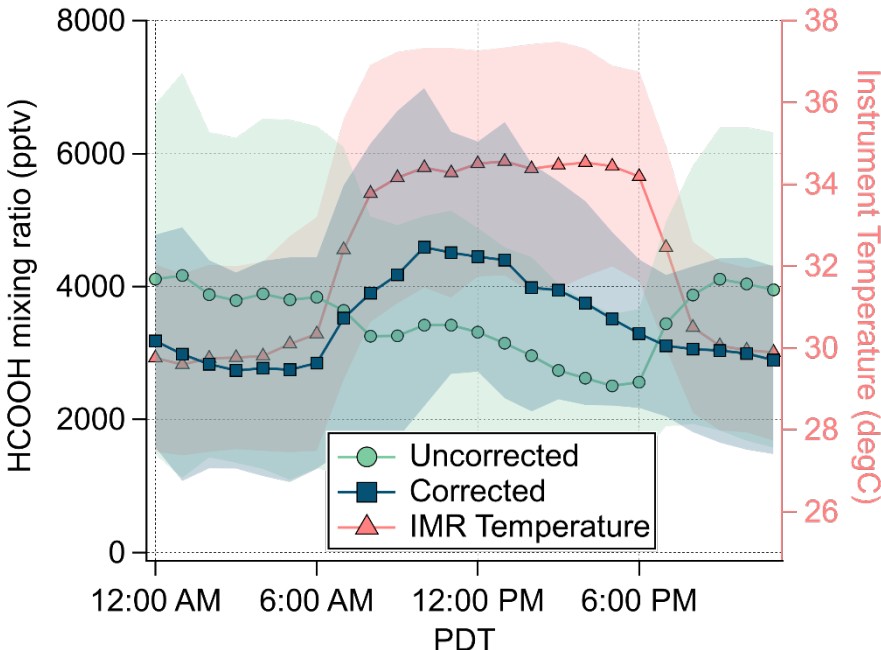

**Figure 5: Campaign average diurnal mixing ratio of HCOOH (uncorrected and corrected for IMR temperature) and IMR temperature. Shading represents 1 standard deviation.**

Sufficient energy can de-cluster even the most strongly bound clusters, as shown by this work and SSQ voltage scanning experiments (Lopez-Hilfiker et al., 2016). Therefore, in order to maintain stable sensitivities in the instrument, understanding the locations where declustering occurs is important to implementing efficient IMR temperature control strategies in the field. During the SUNVEx field deployment of the NOAA CIMS in 2021, the instrument was installed at a ground site, where the IMR temperatures ranged from 25 to 40 °C (298 to 313 K) (see Supplemental Figure 3). The temperature changes were caused both by the PAN inlet dissociator that was on only during the day and ambient temperature fluctuations. This temperature range is not atypical for field locations (i.e. mobile lab, ground site trailer, or research aircraft). Hourly HCOOH standard additions indicated a significant change in instrument sensitivity as a function of IMR temperature, shown in Figure 6b. The variations in sensitivity with temperature in lab experiments and field observations are similar, however absolute sensitivities may differ between the two IMR designs (SUNVEx, NOAA IMR, shown in gray in Fig 6b; Lab experiments, ARI IMR, red, salmon and blue points, Fig 6b). Small differences in these slopes indicate IMR temperature laboratory experiments do not fully mimic the behavior of the instrument when deployed in a harsh environment, suggesting that ion-analyte de-clustering occurs not only in the IMR, but further into the instrument, likely in the SSQ or BSQ.





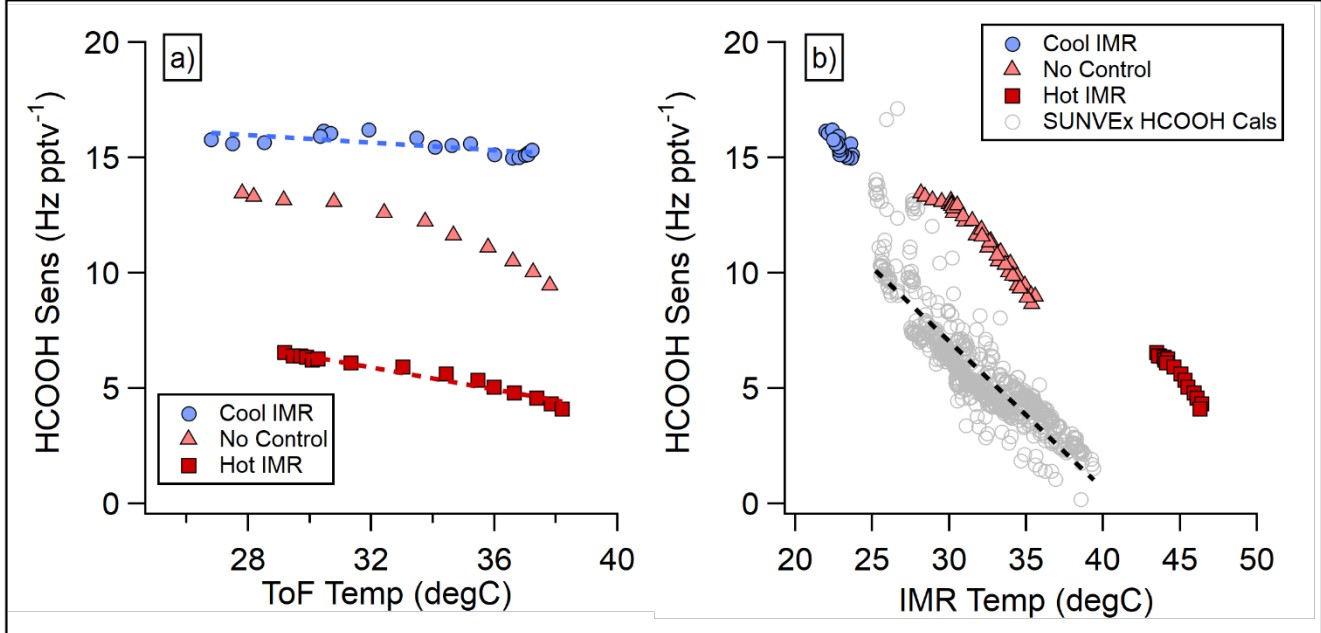

**Figure 6: Comparison of absolute HCOOH sensitivity between laboratory IMR temperature experiments with SUNVEx field standard additions.**

Several ToF body experiments were conducted to determine effective methods for stabilizing sensitivity when environmental temperatures change (Figure 6). Instrument temperatures (described as ToF temperature) varied from 28 to 38 °C over the course of these experiments, as shown in panel 6a. The heated IMR control strategy attempted to maintain the IMR at 45 °C while the ToF temperature varied. The cooled IMR control strategy attempted to maintain the IMR temperature at 18 °C while the ToF temperature varied. Both IMR temperature control strategies improve HCOOH sensitivity stability compared to an uncontrolled instrument. Although, HCOOH sensitivity is reduced with the heated IMR control (red squares), this control strategy still reduces sensitivity drift (~50% reduction) when compared to an uncontrolled IMR (salmon triangles). Cooling the IMR (blue circles) both improves absolute sensitivity and reduces sensitivity drift further (~75% reduction in sensitivity drift) when compared to uncontrolled IMR temperature operation.

## 4 Conclusions

Iodide ion chemistry, commonly used in ToF-CIMS that sample the atmosphere, imparts a temperature dependence to analyte sensitivity. The sensitivity variation with temperature for some analytes is large (5% °C$^{-1}$ from 36 to 40 °C). Importantly sensitivity can be largely controlled by temperature stabilizing the IMR. Further, the sensitivity temperature dependence gives insights into the rates of ion molecule reactions that control the product ion abundance, and provides information on absolute sensitivity. Measuring the temperature dependence of the sensitivity allows for the determination of iodide analyte reaction enthalpies. These enthalpies reveal the reactions that control the abundance of product ions in the IMR. Weakly bound clusters



are strongly temperature dependent and can be detected more sensitively at lower IMR temperatures. This work was prompted by the regular addition of a weakly bound analyte (HCOOH), which has a sensitivity temperature dependence similar to other

weakly bound analytes. We recommend a similar regular calibration approach is taken for all CIMS field deployments. These considerations are likely not unique to the iodide ion adduct system but are more generally applicable to other CIMS ion chemistries. We recommend careful examination of IMR temperature dependencies of instrument sensitivities, especially for weakly bound clusters (many organics) where the effect may be large. Temperature control of the IMR region can help reduce the impact, but de-clustering can occur further in the instrument ion optics. Future work on expanding the range of analyte

classes (for example highly functionalized organics) and ion chemistries will be of interest to improve our skill at predicting CIMS absolute sensitivity.

**Data availability**

Data from the SUNVEx campaign are available to the general public at https://csl.noaa.gov/groups/csl7/measurements/2021sunvex/GroundLA/DataDownload/

**Author contribution**

MR, AN and PV designed the experiments, MR and AN carried them out. MR, AN and PV executed the field project. MR and AN prepared the manuscript with contributions from all co-authors.

**Acknowledgements**

The NOAA Chemical Sciences Laboratory acknowledges support for this work from the California Air Resources Board under
agreement number 20RD002. This work was supported in part by the NOAA Cooperative Agreement with CIRES, NA17OAR4320101.

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
