# Peer review of "Temperature dependent sensitivity of iodide chemical ionization mass spectrometers"

_Atmospheric Measurement Techniques, 2022_

## Author Comment (AC1)

We thank the reviewer for constructive comments which have improved the paper. The reviewer comments are shown below in black. Our responses are shown in blue, and changes to the manuscript are shown in red.

This paper, *Temperature dependent sensitivity of iodide chemical ionization mass spectrometers*, describes observations of instrument performance made during laboratory experiments and previous field campaigns to understand unexpected instrumental behavior. Iodide Chemical Ionization Mass Spectrometers (I-CIMS) are used by numerous groups worldwide to measure a variety of species including halogens, NOZ constituents and oxidized organic compounds. This makes this paper highly relevant to the community.

On its face, it seems obvious that higher temperatures would results in less strongly bound clusters falling apart but it is not something you typically see considered in these types of measurements. Since cluster chemistry is important to other ionization schemes beyond iodide, the significance of this work extends to other ion chemistries as well (eg. NH4+, Br-, Fluoride transfer). One part I feel should be made clearer is which experiments were done with which IMR. I realize the IMR heating experiments were performed with the Aerodyne IMR, but what about the cooling experiments or the room temperature sweep experiments? This is not clear to me and left me confused as to which IMR was used where. I suspect I would not be the only one. I am wondering if the authors could also provide a few more details about the closed loop humidity control. This should be added to the SI. It would give the reader a better understanding of how this system works and the variability of the amount of humidified nitrogen added. Since the I(H2O)- cluster is both a product ion and also a reagent ion the temperature dependence of it has important direct and indirect implications on the instrument sensitivity. The manuscript is well written and falls within the scope of AMT. It should be published once the few minor comments are addressed.

We have added the following sentences to sections 2 and 3.3 in order to clarify which IMR was used for each experiment.

L146:

"The NOAA IMR design was deployed during SUNVEx and the ARI IMR design was used to investigate the influence of IMR temperature in the laboratory."

L169:
"Three types of experiments were performed using the ARI IMR design to examine temperature control strategies that could be implemented to stabilize sensitivity if ambient temperatures change substantially."

L180:
"This experiment replicates conditions during prior field deployments and reveals the sensitivity changes that can occur in the absence of IMR temperature control and was conducted on both IMR designs."

L266:
"The impact on HCOOH sensitivity due to IMR temperature changes during the SUNVEx field campaign (NOAA IMR) are shown in Figure 5."

L282:
"The variations in sensitivity with temperature in lab experiments and field observations are similar, however absolute sensitivities may differ between the two IMR designs (SUNVEx, NOAA IMR, shown in gray in Fig 6b; Lab experiments, ARI IMR, red, salmon and blue points, Fig 6b)."

L291:
"Several ToF body experiments were conducted on the ARI IMR to determine effective methods for stabilizing sensitivity when environmental temperatures change (Figure 6)."

Additionally we have added a section to the supplemental material to describe our IH2O control strategy more clearly L134:

"A detailed description of the CR closed loop control system can be found in the supplemental section 1."

**1 Ion molecule reactor cluster ratio closed loop control system**
The NOAA I- chemical ionization mass spectrometer (CIMS) utilizes a closed loop control system to achieve fixed cluster ratio (I–•H2O:I–) (CR) in the ion molecule reactor (IMR) with varying sample gas humidity. This closed loop control system is comprised of a water bubbler, N2 mass flow controller (MFC), saturated gas transfer line and computer control software.  This system is outlined in supplemental figure 1. Real time signal measured at nominal mass 127 (I–) and 145 (I–•H2O) is transferred from TofDAQ to the National Instruments Labview instrument control software.  The apparent CR is determined at 0.1 Hz and compared to the user defined reference CR set point (typically 0.5), an error is determined and the saturated N2 flow (0 – 100 sccm) MFC set point is adjusted accordingly. This system dynamically adjusts the amount of saturated N2 flow delivered to the IMR as inlet sample gas humidity changes with altitude (on an aircraft) or time of day (at a ground site).

[Figure]

**Supplemental Figure 1: Block diagram of closed looped control of IMR cluster ratio.**

Specific Comments
P1 L 23: Sentence should be reworded. If gives the reader the impression that only cooling reduces the sensitivity drift.

"Finally, we recommend two approaches to minimizing this effect in the field, namely heating or cooling the IMR; the latter has the added benefit of improving absolute sensitivity and reducing drift in harsh field environments."

Now reads

"Finally, we recommend two approaches to minimizing this effect in the field, namely heating or cooling the IMR; the latter has the added benefit of improving absolute sensitivity."

P1 L26: Atmospheric trace….
Corrected.

P3 L73: This is a very good point. I am struggling to think any temperature dependent studies beyond some of the initial kinetics papers in 90's.
We agree.

P4 L103: The authors should provide a couple of example analytes not merely the reference.
This sentence now reads:
"Temperature affects the abundance of product ions, and hence the sensitivity of iodide adduct CIMS, as has been shown for some analytes studied in the laboratory, such as the iodide carboxylate anion reaction (R3) and the Br–(HO2) cluster (Villalta and Howard, 1996; Sanchez et al., 2016)."

P4 L114: The stabilization of sensitivity assumes the clusters make it the detector without falling apart, also dependent on the operating pressure of SSQ there is potential for chemistry to continue into this region which is never temperature controlled on any of these instruments.

We agree with the reviewer that declustering in the SSQ should have an impact on sensitivity, however temperature controlling the entire instrument was deemed out of scope for this work. We believe that most users can benefit by temperature controlling their IMR system, greatly improving the repeatability of in-field calibrations. We have reworded L114:

"Consequently, we compare our experiments to field observations and recommend temperature control strategies to improve sensitivity stability."

P5 Figure 1: It seems odd to me that the nylon liner on the NOAA IMR does not provide some form of temperature isolation from the outside temperature of the stainless steel fittings. Clearly, it's just not enough isolation

Experiments were done before the SUNVEx campaign in attempt to isolate the issue, and we determined the nylon sleeve does not have a significant impact on HCOOH sensitivity dependence on temperature. We have chosen to not include these experiments in this manuscript.

P5 L131: How much N2 do you have to flow through the bubbler? Is it the same flow to maintain the same cluster I-:I(H2O)- cluster distribution in the two IMR's? I am curious because dependent on IMR geometry I have seen this vary greatly in our instrument.

We flow 0 to 100 sccm of $N_2$ flow through our bubbler to maintain cluster ratio. We have updated L131 to read:

"These experiments (unless stated otherwise) were done with water dynamically added to the IMR (via a saturated small N2 flow (0 – 100 sccm)) to maintain a cluster ratio (I–•H2O:I–) (CR) of $0.50 \pm 0.002$, as is typical in field operation of the instrument."

We see similar cluster distribution effects due to IMR geometry in our instrument. During a ToF Temperature experiment, the NOAA IMR design requires 15 sccm of saturated $N_2$ flow to maintain a CR = 0.5 at 28 °C and 19 sccm of saturated $N_2$ flow to maintain that same CR at 30 °C. The same experiment conducted with the ARI IMR design requires 5 sccm of saturated $N_2$ flow to maintain a CR = 0.5 at 28 °C and 7 sccm of saturated $N_2$ flow to maintain the same CR at 30 °C.

We have included an additional figure in the SI (Supplemental Figure 3) to show this difference in designs:

[Figure]

**Supplemental Figure 3: Comparison of NOAA IMR and ARI IMR during ToF body temperature experiments.**

And have added L179:

"A comparison of IMR saturated N2 flow rates during the ToF body temperature experiment between the two investigated IMR designs is provided in Supplemental Figure 3. The changes observed in added IMR saturated N2 flow is likely due to geometry differences between designs and the impact on mixing in the IMR."

P5 L 138: I'm curious why the authors chose only to normalize by the I(H2O)- cluster. Granted it should account for any variation in reagent signal it is merely that often in the literature the reagent is taken as the sum of the two.

The primary reason is that for many species of interest to us (PAN, HNO3, formic, halogens) the reaction proceeds almost entirely with I(H2O)- as the regent ion. Therefore normalization using IH2O- most accurately accounts for changes in the primary ions. However, for species that react strongly with I- as the primary reagent ion (HCN for example), normalization to I- is the more appropriate choice. Further as we show in this work several species react with both IH2O- and I- so the sum may be more appropriate. Ultimately the choice of normalization is somewhat arbitrary as long as it is held consistent throughout calibration and analysis. However,

normalization using IH2O- of a species that reacts predominantly with I- (again HCN for example) can impart an apparent 'humidity dependence' and should be considered. It is our not so satisfactory conclusion is that normalization should be a compound specific decision, but careful processing of data and laboratory calibration can minimize the potential for error. It is worth stating that as we show in this work a given CR may not be identical across instruments so caution should be used even when comparing normalized sensitivities between instruments.

P6 L157: From the looks of the temperature profile these experiments, they were not done as ramp and soak type experiments. I'm curious as why this decision was made as opposed to trying to assess the issue at discrete temperatures. Granted this mimics what typically occurs in most measurement trailers or aircraft.

These experiments were designed to mimic SUNVEx trailer temperature profiles fit into several hours rather than the 24 hrs.

P7 L172: Do you have any idea as to the actual gas temperature in the IMR's? I am guessing even with heating or cooling they would be very different when the instrument is operated as a PAN CIMS.

We have conducted experiments which indicate the IMR excess flow temperature is very similar to the IMR wall temperature (~1.5 °C offset) both in PAN CIMS mode and normal operation. We have not conducted these experiments for heating or cooling the IMR yet, but plan to once we have designed and fabricated a new IMR.

P7 L179: are shown

Corrected.

P8 Table 1: I am curious why you did not calculate the sensitivities for 2-nitrophenol and phenol if you know what mixing ratio added to the inlet is?

These experiments do not require us to know the inlet mixing ratio, only that the inlet mixing ratio is stable during the experiment. At the time of these experiments, we did not have the ability to calibrate for 2-nitrophenol or phenol.

P8 L204: It is not exactly fair to compare the PAN chemistry (particularly the anion chemistry) when the ionization chemistry is a different mechanism than the rest. The authors should specify which channel or drop it and merely discuss the halogens.

The authors agree that the anion PAN chemistry differs from the other analyte clustering chemistries discussed here. However, we believe it is still interesting to the reader and useful to PAN CIMS users, to put PAN temperature dependence into context of other iodide CIMS

analytes.  We believe we have made the mechanism clear by indicating a reaction number on L204:

"Notably the sensitivity of several halogens remain nearly constant with temperature and the PAN carboxylate anion (R3) sensitivity does not decrease with temperature."

P9 L212: Again, I am not sure grouping the PAN anion channel with the others is appropriate.

In order to make the grouping clearer, we have reworded L212 as follows:

"The analytes studied here can be grouped into two classes, strongly bound clusters with weak IMR temperature dependence (such as $Br_2$, $N_2O_5$, and $ClNO_2$) and weakly bound clusters with strong IMR temperature dependence (HCl, HONO, HCOOH, HCN and PAN via peroxy radical cluster). The carboxylate anion detection of PAN behaves similarly to strongly bound clusters, but is not clustering ion-molecule chemistry and cannot be classified."

P9 L215: Why were the sensitivities not determined?

At the time of these experiments, we did not have the ability to calibrate for 2-nitrophenol or phenol.

P9 L227: How much nitrogen needs to be added/removed for the humidity control? Are we talking about 10's of sscm on a couple of litres or is it more? I am curious if there is any possibility of dilution in the IMR and without that information, it is not clear to the reader.

During normal operation 10s of sccm are added to the IMR to control cluster ratio.  As we have found here, this increases with increasing temperature due to declustering of the IH2O cluster. We have corrected for this possible dilution artifact, and the figures and slopes presented in Table 1 have not changed appreciably.  We have added the text L131:

"The added saturated $N_2$ flow can dilute the sample in the IMR and is typically a negligible effect under stable temperature operating conditions, but due to the nature of these experiments (large $N_2$ flow at high IMR temperatures) we have corrected for dilution."

P10 L230: I(H2O)- is also a reagent ion. If it is falling apart in the IMR as a results of increasing temperature that would have a direct effect on sensitivity beyond clusters falling apart downstream.

We agree with the reviewer on this topic and have reworded this sentence to emphasize the importance of controlling cluster distribution and IMR temperature, as they can both impact sensitivity in concert L235:

"This thermodynamically controlled behavior of the I–•(H2O) product ion, which is also a critical reagent ion in our IMR, emphasizes the importance of temperature and CR control to maintain a constant reagent ion distribution and stable sensitivity."

P12 L280: This is definitely true.
We have adjusted the text to read L286:

"The variations in sensitivity with temperature in lab experiments and field observations are similar, however absolute sensitivities differ between the two IMR designs (SUNVEx, NOAA IMR, shown in gray in Fig 6b; Lab experiments, ARI IMR, red, salmon and blue points, Fig 6b)."

P12 L283: At what pressure do you operate the NOAA I-CIMS SSQ? 2 mBar or lower?

We have adjusted the text on L119 to indicate our SSQ operating pressure:

"The NOAA CIMS notably operates at lower IMR pressure (40 mbar, rather than 100 mbar used in many) and small segmented quadrupole (SSQ) pressure (1.64 mbar, rather than 2 mbar) in order to reduce the impact of secondary chemistry in the IMR and SSQ instruments (Bertram et al., 2011; Lee et al., 2014)."

P13 L294: Are the sensitivities the same using both NOAA IMR and the Aerodyne one? There are definitely geometry differences between the two. Were the cooling experiments only done with the NOAA IMR and heating experiments only done with the Aerodyne IMR?

We point the reviewer to Figure 6b, where absolute sensitivities are reported for HCOOH for both IMR designs. We have adjusted the text in the figure caption to make this comparison more clear L294:

"**Figure 6: Comparison of absolute HCOOH sensitivity between laboratory IMR temperature experiments (ARI IMR design) with SUNVEx field standard additions (NOAA IMR design)."**

---

## Author Comment (AC2)

We thank the reviewer for the thorough and thoughtful review. The reviewer comments are shown below in black. Our responses are shown in blue, and changes to the manuscript are shown in red. We have reworded several sections of the manuscript and reorganized the introduction to make it clearer. We have added a figure to the SI, which shows instrument temperature data from a research aircraft campaign, moved experimental example figures from the SI to the main text and included an example of the Van't Hoff fit in the SI to add clarity to Section 3. Additional details of each IMR design are also included in Supplement Table S1.

This publication details a set of laboratory experiments examining the effect of temperature on a chemical ionization reactor employing the iodide anion as a reagent. The experimental results are then used to correct a field data set for ambient temperature variations.

In some ways this is an unusual manuscript. It contains information that typically would comprise the SI of a paper detailing the results of the ambient measurements. But I think that hiding many of these experimental instrumental details in the SI has previously been a disservice to the community. SI sections are rarely read nor widely disseminated, and such important details should be out in the open as these instruments proliferate across atmospheric chemistry.

However, this manuscript should be more clearly written and organized than it currently is.

I support publication eventually after the authors address the items listed below. More experiments should not be necessary, but I suggest major rewrites for clarity. Additionally, there are several places that the authors speculate about instrumental details and results without supporting evidence. These should be clarified with much more precise language.

**Specific Comments**

L87: "This discrepancy…" The authors should add proof of this speculation via a citation or data.

We agree this sentence was speculative and have removed it from the manuscript.

L95: The connection between this work and Lopez-Hilfiker's voltage scanning method is not clear to me, nor is the reason for this paragraph.

The voltage scanning method and our work is connected by well understood drift tube studies which show that the drift of reagent ions in an electric field increases both the ion gas energy and the ion-molecule interaction energy, akin to increasing temperature in the drift tube (Spesyvyi et al., 2015). Voltage scanning explores how the strength of the electric field between the back of the SSQ (Skimmer voltage) and front of the BSQ impacts declustering of product ions detected

by the instrument. Our work explores how certain product ion clusters decluster due to increases in kinetic energy (i.e. temperature), significantly influencing instrument sensitivity. We have reworded this paragraph to make this connection clearer to the reader:

"Analyte sensitivity in CIMS is a product of the net rate of product ion formation, a function of adduct binding energy, and the instrument transmission of those adduct ions. A recent framework for understanding iodide adduct binding energies uses a mass spectrometer ion lens voltage scanning procedure to estimate the relative binding energies (Lopez-Hilfiker et al., 2016). This approach gives important insight to the stability of ion adducts and the extent to which they may decluster during transmission, and can be used to provide inferred or relative sensitivities (Lopez-Hilfiker et al., 2016). A drawback of the voltage scanning method for determination of relative adduct binding energies is the potential for unintended changes in instrument ion transmission, which may result from modulation of instrument tuning to induce the needed change in field strength. In this work we leverage kinetic energy in the form of a temperature change as a substitute for modulation of the field strength, thereby eliminating the impact of ion transmission. The effect of temperature on instrument sensitivity has not been widely explored for the iodide adduct CIMS, where the impact has been documented for limited analytes such as the iodide carboxylate anion reaction (R3) and the Br–(HO2) cluster (Villalta and Howard, 1996; Sanchez et al., 2016)."

L108: Typically the last paragraph of the introduction focuses on what the forthcoming paper has done. Instead, this manuscript introduces another paragraph referencing previous temperature control strategies. This paragraph is disorienting. It should be moved to earlier in the introduction or later in the results. Also, the authors reference the FIGAERO as an IMR temperature control strategy, which is incorrect. The FIGAERO is an inlet that goes on the front of the IMR and is independent of the IMR itself, separate from the IMR's temperature regulation or lack thereof.

We agree with the reviewer that the order of paragraphs in the introduction could be made clearer and have reorganized this section of the manuscript. The last two paragraphs of the introduction now read:

"Previously, several CIMS IMR temperature control methods have been implemented. Documented instrument deployments range from actively heating the IMR region or hardware close to the IMR (50 to 60 °C), such as for the Filter Inlet for Gases and Aerosols instrument (Lopez-Hilfiker et al., 2014), to insulation of the IMR region (Lee et al., 2018), to active cooling of the IMR region (15 °C) (Neuman et al., 2002). However, thermal coupling between the IMR and ToF body, which may be difficult to temperature control in field deployments of CIMS instruments (discussed below), may introduce temperature variations in regions where ion molecule reactions occur.

We extend the impact of temperature on the abundance of product ions to a range of atmospheric trace gases and provide a framework for understanding variations of detection sensitivity with temperature for I– ion molecule reactions. Additionally we estimate thermally controlled reaction pathways in our IMR system by comparing observed reaction enthalpies to literature values.

Finally, we compare our experiments to field observations and recommend temperature control strategies to improve sensitivity stability."

L117: My understanding of the Tofwerk instrument is that it is an OEM instrument, onto which others can install ionization sources. So this would seem to not be a "modified commercially available TOF", which the authors directly contradict anyway in L123 where they say the "TOF has not been modified."

Line 117 refers to a "modified commercially available ToF-MS", i.e., referring to the instrument as a whole rather than just the time of flight (ToF) region as the reviewer suggests. The NOAA CIMS has been modified for aircraft deployments, which required a custom pressure control region before the IMR. Additional modifications include, additional pumping capacity and pressure control systems to run our IMR and SSQ at lower pressures than the OEM instrument and a cluster ratio control system which greatly improves the stability of analyte sensitivity. However, as the reviewer indicates, we have not modified the ToF region of the instrument itself, so for consistency with L123, we have adjusted the text on L117 to read:

"…modified commercially available CIMS…"

L120 seems to refer to the IMR, but these running conditions in Lee et al. are specific to Iodide and Bertram et al. used pressures of 20-100 mbar. This entire paragraph should be edited for clarity.

Bertram et al. outlines the first deployment of a chemical ionization TOFMS made by Tofwerk and Aerodyne Inc, granted using acetate ions, but encourage the reader to consider using other common CIMS reagent ions (such as $I^-$, $CF_3O^-$, $SF_6^-$). Bertram et al. clearly state that the IMR pressure is typically operated at 85 mbar, and the pressure range of 20 to 130 mbar is used for diagnostic experiments. We reference Lee et al.'s 90 mbar IMR operating condition, as this paper is often considered a standard for current Iodide CIMS implementation. Our instrument, an Iodide CIMS, notably runs at a lower operating pressure of 40 mbar. We have rephrased the parenthetical statement to include the word "instrument" for clarity, and removed the word "instrument" after IMR on line 121:

"The NOAA CIMS notably operates at lower IMR pressure (40 mbar, rather than 100 mbar used in many instruments) and small segmented quadrupole (SSQ) pressure (1.64 mbar, rather than 2 mbar) in order to reduce the impact of secondary chemistry in the IMR and SSQ (Bertram et al., 2011; Lee et al., 2014)."

L121 "It is worth noting that higher pressure IMR systems will likely be more susceptible to thermal effects due to the increased residence time" certainly seems possible but is still speculative and should be supported in some way.

We agree with the reviewer that this sentence at this point in the manuscript may seem speculative and have moved the sentence to the conclusion of the manuscript L326:

"It is worth noting that higher pressure IMR systems may be more susceptible to thermal effects due to the increased time to reach equilibrium."

Our work shows that analytes that reach equilibrium with the reagent ion on the time scale of residence in the IMR will exhibit temperature dependence in sensitivity. Therefore we believe it is not a speculative statement to propose IMRs with longer residence times (i.e. higher operating pressure) will exhibit more ion-molecule reactions reaching equilibrium than lower pressure systems.

L140 I was surprised here to read that this work also uses the ARI IMR since the entire introduction focuses on the NOAA-built ionization source. There is no prior introduction to the ARI IMR and no discussion on its specifics, nor a citation where it's referenced here. I suggest adding more detail on the ARI IMR to the main methods section (more than just the tiny table in the SI). Further, regarding the L117 comment, is the NOAA instrument any different than the commercially modified one if it sometimes employs the ARI IMR?

We choose to use the ARI IMR in order to show the community that this impact occurs on any non-temperature controlled IMR in which the ion-molecule reaction system is allowed to come to equilibrium on the time scale of IMR residence times. We point the reviewer to L143 where the two IMR designs used in this work are introduced. We have added appropriate references for the ARI IMR on L154:

"Experiments were conducted using the widely used temperature controllable ARI IMR, which includes a cartridge heater in a stainless steel block mounted to the IMR and controlled via the instrument computer system (Bertram et al., 2011; Krechmer et al., 2016)."

Additionally we have added IMR volumes and residence times at 40 mbar to Supplement Table S1:

| IMR Design | Materials | Temperature control range (°C) | Residence Time (40 mbar) (ms) | Volume (cm$^3$) | Image |
|---|---|---|---|---|---|
| ARI | Stainless Steel + PEEK (non-wetted) | 30 – 50 | 46 | 47 |  |

| NOAA | Stainless Steel and Nylon | Ambient temperature | 42 | 39 |  |
|---|---|---|---|---|---|

We point the reviewer to our response above regarding the differences between the commercially available API-TOF and the NOAA CIMS.

Section 2.1: The authors start this paragraph off making a claim that a constant IMR but a changing TOF temperature would cause temperature-dependent ion chemistry in focusing ion optics. But it doesn't appear that the authors conducted this experiment. Could they please support this claim?

We respectfully disagree that the experiment in question has not been conducted and point the reviewer to Figure 6. Here we conduct two experiments which point to this effect, a chilled IMR (18 degC) and a heated IMR (45 degC). Each experiment shows a degree of declustering of the I•HCOOH⁻ product ion in the instrument, however we have not determined exactly where this is occurring, but can eliminate the IMR portion of the instrument if the IMR temperature maintains constant. The degree of declustering of this product ion differs between experiments, but taking the better temperature controlled case of cooling the IMR, the declustering effect is minimal. We have adjusted the beginning of this paragraph to read:

"As discussed in section 3.3 below, temperature dependent ion chemistry may occur downstream from the IMR, in which case IMR temperature alone does not determine sensitivity."

Supplemental figures 1 and 2 should be cleaned up and moved to the main paper as a multi-panel figure referenced in Sections 2.2 and 2.3. This is a short paper and its entire value is that it is not hidden in the SI (see general comment above).

We agree that it is useful to include an experiment representative time series in section 2. We have included the following figure as figure 2:

[Figure]

**Figure 2: Typical time series for an IMR temperature experiment. Product ion signal is normalized to I⁻•H₂O signal.**

L239: The text references slopes on the graph, which I think is a good idea, but there are no slopes. The authors should add slopes to Figure 3, or if too busy, add a plot with an example regression. I find section 3.2 to be the most interesting, but it is short on details and specifics.

The slopes of Figure 3 are tabulated as the Y-axis of Figure 4 as net reaction enthalpies. The authors found adding slopes to Figure 3 was too busy, so did not include them. However, we have included an example regression as Supplement Figure S4:

[Figure]

**Figure S4: Example Van't Hoff relationship fit for HCN. The slope of this fit represents** $-\frac{\Delta_r H^0}{RT}$ **and is a measure of net reaction enthalpy under the conditions of the IMR.**

L258: I'm skeptical that HCl "fall[s] close to the 1:1 line". It has essentially the same measured reaction enthalpy as Phenol and HCN within the margin of error, but a different literature enthalpies. Is this a small influence of ligand switching? Why does nitrophenol fall so far off the line and is not discussed other than having a low overall sensitivity? This section could benefit from more analysis and textual interpretation.

We have not included the literature (or NOAA CIMS experimental) uncertainties in this figure. However the literature reported uncertainty for HCl clustering reaction with iodide is quite high, 61.9 ± 8.4 kJ/mol. NOAA CIMS measured net reaction enthalpy is estimated to be 69.4 ± 0.9 kJ/mol making this comparison quite favorable. We found including measurement uncertainties to make this figure too busy for interpretation, however we have adjusted L258 to read:

"These analytes differ from HCN, HCl and phenol which, within experimental uncertainty, all fall close to the 1:1 line for the direct clustering pathway (see Figure 4). For example, the literature reported value for the HCl clustering reaction is -61.9 ± 8.4 kJ/mol which compares well within the experimentally determined value of -69.4 ± 0.9 kJ/mol."

2-Nitrophenol does not fall far from the 1:1 line, but straddles it similar to HCOOH, possibly indicating the thermochemical system lies close to the ligand switching net reaction enthalpy. Experimental uncertainties are again quite high in the literature with a reported clustering reaction enthalpy of 97.1 ± 7.5 kJ/mol. We have adjusted the text on L266 to make these uncertainties and where 2-nitrophenol thermochemical system falls more clear:

"A similar conclusion can be drawn for 2-nitrophenol, as the measured net reaction enthalpy of -58.6 ± 0.6 kJ/mol falls between the reported ligand switching reaction enthalpy of -54.1 ± 8.1 kJ/mol and direct clustering reaction enthalpy of -97.1 ± 7.5 kJ/mol."

L277: These field temperature swing would be a useful SI figure and the authors should have plenty of good examples.

We point the reviewer to Supplemental Figure 3 for a ground site trailer example. We have not deployed the NOAA CIMS in a mobile laboratory, so cannot provide this data example. We have added a research aircraft data example as Supplemental Figure 6:

[Figure]

**Supplemental Figure 6: Example NOAA CIMS instrument temperature (ToF body temperature) from six research flights (color traces) during the FIREX-AQ research aircraft deployment in 2019. Black trace and shading represent the average ± 2 standard deviations.**

L308: "Temperature control of the IMR region can help reduce the impact, but de-clustering can occur further in the instrument ion optics." The link between collisional fragmentation and temperature dependence is not clear to me as this is written.

The author's intent with this sentence is to warn users that IMR temperature control may not fully eliminate the temperature dependence of iodide CIMS sensitivity, as declustering of the product ion can occur further in the instrument. We have adjusted this sentence to read L308:

"Temperature control of the IMR region can help reduce the impact, but further de-clustering (either due to temperature or collisional fragmentation) of the product ion may occur in the instrument ion optics."

All over: sensitivities in units of Hertz or ions/s should be defined with an extraction frequency for the TOF.

We have added a sentence to describe our ToF's extraction frequency during this work L139:

"Ion packet extraction frequency in the time of flight region of the instrument during this work was 25,000 Hz."

**Technical Comments**

L99: I'm not sure I follow the sentence at L99. Please clarify or add more detail: "Additionally, product ion formation dependence on temperature may make these voltage scanning determinations difficult to interpret"

These sentences have been removed and this section has been reworded for clarity.

L51: "refined restriction policy" – This could be clearer

We do not understand this comment, as "refined restriction policy" does not appear in our manuscript.

L241: The nist webbook should be a citation in common citation format

Corrected.

L299: "dependence of sensitivity on temperature"

Corrected.

**References**

Spesyvyi, A., Smith, D., and Španěl, P.: Selected Ion Flow-Drift Tube Mass Spectrometry: Quantification of Volatile Compounds in Air and Breath, Anal. Chem., 87, 12151–12160, https://doi.org/10.1021/acs.analchem.5b02994, 2015.